METHODS AND RESOURCES

# IVEN: A quantitative tool to describe 3D cell position and neighbourhood reveals architectural changes in FGF4-treated preimplantation embryos

Jessica E. Forsyth[1,2], Ali H. Al-Anbaki[1], Roberto de la Fuente[1,3], Nikkinder Modare[1], Diego Perez-Cortes[1], Isabel Rivera[1], Rowena Seaton Kelly[1], Simon Cotter[2], Berenika Plusa[1]*

**1** Faculty of Biology, Medicine and Health (FBMH), Division of Developmental Biology & Medicine, Michael Smith Building, University of Manchester, Manchester, United Kingdom, **2** School of Mathematics, Alan Turing Building, University of Manchester, Manchester, United Kingdom, **3** Department of Experimental Embryology, Institute of Genetics and Animal Biotechnology, Polish Academy of Sciences, Jastrzębiec, Poland

\* berenika.plusa@manchester.ac.uk

**Data Availability Statement:** Data used to generate figures and analyses are available from the GitHub repository https://github.com/

## Abstract

Architectural changes at the cellular and organism level are integral and necessary to successful development and growth. During mammalian preimplantation development, cells reduce in size and the architecture of the embryo changes significantly. Such changes must be coordinated correctly to ensure continued development of the embryo and, ultimately, a successful pregnancy. However, the nature of such transformations is poorly defined during mammalian preimplantation development. In order to quantitatively describe changes in cell environment and organism architecture, we designed Internal Versus External Neighbourhood (IVEN). IVEN is a user-interactive, open-source pipeline that classifies cells into different populations based on their position and quantifies the number of neighbours of every cell within a dataset in a 3D environment. Through IVEN-driven analyses, we show how transformations in cell environment, defined here as changes in cell neighbourhood, are related to changes in embryo geometry and major developmental events during preimplantation mammalian development. Moreover, we demonstrate that modulation of the FGF pathway alters spatial relations of inner cells and neighbourhood distributions, leading to overall changes in embryo architecture. In conjunction with IVEN-driven analyses, we uncover differences in the dynamic of cell size changes over the preimplantation period and determine that cells within the mammalian embryo initiate growth phase only at the time of implantation.

jessforsyth/forsyth-et-al-2021/. Code for the MATLAB and Python 3 IVEN versions is located on the GitHub repository https://github.com/jessforsyth/IVEN-Code.

**Funding:** JEF was supported by the Wellcome Trust (https://wellcome.org/) 4 year PhD studentship (Quantitative and Biophysical Biology, 108867/Z/15/Z). AHA was supported by a scholarship from Iraqi Cultural Attaché in London (GB) (AL-ANBAKI Ref. S-1007). Work of RF and all experiments in BP's lab were supported by the Wellcome Trust grant Seed Awards in Science (212372/Z/18/Z). The funders had no role in study design, data collection and analysis, decision to publish, or preparation of the manuscript.

**Competing interests:** The authors have declared that no competing interests exist.

**Abbreviations:** BSF, Biological Service Facility; DT, Delaunay triangulation; Epi, Epiblast; ICM, inner cell mass; IVEN, Internal Versus External Neighbourhood; PrE, primitive endoderm; TE, trophectoderm.

## Introduction

Dynamic relations between cells in the changing architecture of developing organisms are an important part of the developmental process, albeit rarely investigated in detail. Cell shape and cell position within the whole embryo or particular tissue are often interlinked and respond to mechanical forces, which, in turn, can modulate gene expression and protein activity via the actin cytoskeleton [1–4]. During early mammalian embryogenesis, after a series of symmetric divisions, cells acquire the ability to sense changes in geometry that occur around the time of compaction. Compaction is a developmental process that results in an increase in cell–cell contacts [5,6], thus allowing cells to integrate signals from different membrane domains to assess their relative position within the embryo [7–9]. This is a crucial step in the lineage specification process. Outside-facing cells develop a polarised apical domain and give rise to the first epithelium, the trophectoderm (TE). The TE is the founder lineage of the conceptus part of the placenta. Apolar cells devoid of the apical domain are internalised and give rise to the inner cell mass (ICM). ICM cells then differentiate further into 2 populations of cells: the Epiblast (Epi), which gives rise to the embryo proper, and the primitive endoderm (PrE), which contributes predominantly to the extraembryonic supportive tissues [10,11]. Maturation of TE allows accumulation of fluid, initially in the form of small vesicles that gradually increase in volume and merge into one large, asymmetrically positioned, fluid-filled cavity that displaces ICM cells towards the opposite region of the embryo [11]. The TE envelops both the ICM and the cavity, and the morphological and geometrical changes related to cavity formation herald the formation of the blastocyst [11,12].

The way in which a local cell neighbourhood changes in response to major changes in embryo architecture remains poorly understood. One of the major roadblocks is a lack of appropriate tools that allow for large-scale, quantitative assessments of the changes in the cell microenvironment and the ability to link them to overall transformations in embryo geometry.

We defined a cell's local environment, or microenvironment through the concept of a "neighbourhood", as in the number of neighbouring cells or neighbours. To enable the quantitative description of a cell's local neighbourhood during major developmental events, we developed the pipeline Internal Versus External Neighbourhood (IVEN). IVEN is a quantification tool that has been implemented as a user-friendly, open-source software that can be used in conjunction with existing open-source image analysis software like MINS [13], Nessys [14], and ImageJ, as well as commercially available programs like IMARIS (Bitplane). Using the IVEN pipeline allows for a quantitative analysis of the spatial cell distribution within an embryo or tissue and links it to the overall embryo architecture.

The use of IVEN allowed us to classify embryonic cells into clearly defined populations (like internal and external cells) as well as to distinguish further subpopulations within each group and quantify changes in cell neighbourhood related to the development of TE as an epithelium. In addition, we were able to quantify the changes in ICM cell neighbourhoods that accompany the blastocyst formation and maturation process, which lead to the formation of the most inner ICM population with very little or no contact with TE cells. IVEN-driven analyses revealed that treatment of mouse embryos with FGF4 leads to changes in cell organisation and neighbourhood distribution that, in turn, are reflected in alteration of the whole ICM geometry. Our results demonstrate that evident changes in cell neighbourhood during development are accompanied by decrease in cell size until the time of implantation.

## Materials and methods

### IVEN development

The IVEN pipeline was developed in MATLAB and Python 3, and all files and tutorials are available at https://github.com/jessforsyth/IVEN-code. The pipeline uses ".xls" files as input and requires data to be compiled into the format outlined in the detailed tutorials. IVEN was designed to make use of inbuilt MATLAB/Python functions and has been written to be easily adjusted by the user but has been primarily designed to accept 3D data. IVEN first classifies cells as outside or inside using a convex hull algorithm. Cells that are used to generate the convex hull are classified as outside cells, whereas cells that are enclosed by the convex hull are classified as inside cells. A user-interactive window is then opened to allow the user to correct the initial automatic cell classification where necessary.

For thoroughness, we also compared the performance of the convex hull algorithm with an ellipsoid fitting technique similar to already existing methods of cell classification. We used a freely available ellipsoid fitting package from the MathWorks File Exchange. Cells were identified as outside cells according to the following scoring method, $v = \left(\frac{x-o_x}{a}\right)^2 + \left(\frac{y-o_y}{b}\right)^2 + \left(\frac{z-o_z}{c}\right)^2$, where $o_x$, $o_y$ and $o_z$ are the fitted ellipsoid centres, and $a$, $b$, and $c$ are the respective ellipsoid radii. If $v$ had a value greater than or equal to 1, then the cell was classified as an outside cell. This approach was deliberately more simplistic than those methods included in [13] to ensure fair comparison to the convex hull approach being tested for use within IVEN.

IVEN calculates the number of neighbours of each cell using a "corrected" Delaunay triangulation (DT), which can be constructed in 3 dimensions [15,16]. The number of neighbours of a cell is inferred from the 3D DT by considering the number of edges from a given cell to other cells. If a cell is "connected" to 4 other cells, we inferred that the cell has 4 neighbours.

However, the DT does not account for cavities within a set of points, such as the cavity within the preimplantation blastocyst. This resulted in some cells across the cavity from each other being considered neighbours as they were connected by a vector in the DT. To correct for this and remove any "untrue" neighbour artefacts, we imposed a threshold on the distance between neighbours, thus creating a "corrected DT" approach. The threshold limits the maximum distance allowed between neighbours and can either be inferred from the data or preset by the user. The inferred thresholding method is calculated automatically within IVEN. The automatic neighbour distance threshold is calculated using the neighbour distance distribution (distance between all cells and their neighbours) for each embryo analysed. The threshold value is then calculated as $P_{75} + (k \times \text{IQR})$, where $P_{75}$ is the 75th percentile, IQR is the interquartile range of the neighbour distances, and $k$ is a user-determined parameter. This approach can be manipulated further to calculate a threshold for inside cells and outside cells separately. Separate thresholds for inside and outside cells were used for embryos $\geq$32-cell stage to account for stretching of the TE cells. To apply the threshold between an outside cell and another outside cell, the TE threshold was applied and vice versa for inside cells. For cases where an inside cell is a neighbour of an outside cell, the inside threshold was applied. All analyses presented within this study use this method of thresholding with $k$ equal to 0.5.

### Implementation

All runs of IVEN were performed on a system comprised of an Intel(R) Core(TM) i7-7560U CPU processor at 2.4 GHz, with 16.0 GB of RAM. Both versions of the IVEN pipeline were designed to be used on standard laboratory or personal computers, but for large numbers of cells, the Python 3 version is suggested, as the MATLAB user interfaces take longer to render

when numbers of cells are higher. We also include a MATLAB script to facilitate batch processing of datasets using default parameters defined by the user.

## Embryo collection

Mice were kept under a 12-hour light cycle at the Biological Service Facility (BSF) at the University of Manchester. Strains used were either CD-1 outbred (Jackson Laboratories) or CAG:: H2B-EGFP transgenic mice to enable visualisation of the chromatin [17]. Presence of a vaginal plug indicated mating of mice and the midday of the plug observance was scored as embryonic day 0.5 (E0.5) of development. Embryos were collected by flushing of the oviducts (up to E2.5) or the uterus horns (E3.5 and later) with warmed M2 medium [18]. Embryos that were not subsequently cultured ex vivo had the zona pellucida removed using acid Tyrode's solution (Sigma Aldrich). Embryos were moved between 2 and 3 drops of warmed solution and visually inspected until the zona was fully removed. After zona removal, embryos were washed and left to recover in M2 for a minimum of 20 minutes. Mouse handling and husbandry followed the regulations established in the UK Home Office's Animals (Scientific Procedures) Act 1986. The animals were bred on project license P08B76E2B, protocol 4 and the license 70/08858, protocol 4. All animals were humanely euthanised according to Schedule 1 of the UK Animals (Scientific Procedures) Act 1986. Ethical approval for the euthanasia of animals for use in this study was granted to the project submitted by Berenika Plusa by the University of Manchester Animal Welfare and Ethical Review Body on the 10/03/2017.

## Embryo fixation and immunostaining

After zona removal, freshly flushed embryos were then fixed in 4% para-formaldehyde (Sigma Aldrich) in PBS with 0.1% Tween-20 (Sigma) and 0.01% Triton X-100 (Fluka) for 20 minutes. Fixed embryos were then washed and stored in PBS at 4˚C.

Immunostaining was performed using the protocol outlined in [19]. All embryos were permeabilised in 0.65% Triton-X 100 in PBS for 20 minutes and blocked in 10% donkey serum (Sigma Aldrich) in PBS for 40 minutes. Primary antibodies used were anti-CDX2 (BioGenex) at 1:1, anti-E-cadherin (Santa Cruz Biotechnology) at 1:500, anti-Nanog (Cosmo Bio, R&D Systems) at 1:300/200, and anti-Gata4 (Santa Cruz Biotechnology) at 1:200 overnight at 4˚C. Secondary Alexa Fluor (Invitrogen) conjugated antibodies were used with a dilution of 1:500 in blocking buffer for 1 hour at 4˚C. To visualise the nuclei, Hoechst 33342 (Sigma Aldrich) was used at a concentration of 1:1,000 in PBX (PBS +0.1% Triton-X 100) and incubated for a minimum of 30 minutes at 4˚C.

## Embryo disaggregation

Embryos were disaggregated using a large reservoir of M2 calcium-free medium and then transferred into an M2 calcium-free drop under mineral oil (Sigma). The embryos were moved between drops of calcium-free M2 and were kept individually in drops to ensure no mixing of cells between embryos. Embryos were gently pipetted using blunted glass pipettes. Once disaggregated, cells were transferred to drops of FM4-64FX dye (1:200 in M2) (Thermo-Fischer Scientific) on a glass-bottomed dish (MatTek).

To disaggregate later-stage embryos (some E4.5 and all E5.5 embryos), embryos were subject to a trypsin–EDTA solution (Sigma, Cat. No. T3924; incubation at 37˚C) for no longer than 5 minutes while embryos were visually inspected for signs of disaggregation. Embryos were then washed in M2 and transferred to microdrops of FM4-64FX dye diluted with M2 (1:200). To ensure that the 2 protocols did not affect cells differently, groups of E4.5 embryos

were disaggregated using both the calcium-free treatment and the trypsin treatment, with no significant differences observed in cell diameters across the 2 treatments.

## Embryo culture

Embryos were cultured in homemade KSOM under mineral oil at 37.5°C and 5% $CO_2$ in air [18]. In order to induce complete PrE composition of the ICM, 1 μg/ml recombinant human FGF4 (R&D Systems) and 1 μg/ml heparin sulphate (Sigma Aldrich) were added to the KSOM media [20].

## Image acquisition

Fixed and stained embryos were imaged on glass-bottom dishes under mineral oil using the Nikon A1 inverted confocal microscope. Sections were imaged every micrometer to provide sufficient spatial resolution for nuclei detection. Lasers used to excite fluorophores were Diode 405 nm, Argon 488 nm, HeNe 546 nm, and HeNe 647 nm.

## Data and image analysis

All images were analysed using IMARIS (Bitplane) and ImageJ. Cell centres were approximated as nuclear centres and identified using the spot detector in IMARIS. The number of cell centres detected was used as the cell count for the embryo to indicate the stage of development. Selection of nuclei centres was checked manually and data exported as Excel workbooks and compiled using an in-house macro. The format of the compiled workbooks allowed for direct import into our program IVEN.

Cell protein intensity (namely CDX2 intensity) was exported using the nuclei centres identified in the IMARIS software. The mean intensity values within each nucleus were exported and subsequently adjusted to account for signal attenuation due to sample thickness, and then normalised per embryo. Intensity values were corrected for signal attenuation by division by nucleus signal intensity (Hoechst). Protein intensity values were then normalised against the maximum signal intensity present within each embryo.

Disaggregated cell diameters were measured using ImageJ. The z slice with the maximum cell cross section was identified and the major and minor axes of the cell measured using the FM4-64FX staining.

## Statistics and data analysis

All graphs presented within this study display the median and interquartile range as error bars, unless otherwise stated. All statistical analyses were conducted using GraphPad Prism. Data were tested for normality using the Anderson–Darling test where samples were sufficiently large, otherwise they were tested using the Shapiro–Wilk test. Where data were distributed normally, a standard ANOVA test was used with subsequent multiple comparison analysis. If data were not normally distributed, the Kruskal–Wallis test was applied with subsequent multiple comparison tests. If the comparison made was between 2 groups and they were nonnormally distributed, the Kolmogorov–Smirnov test was used, or if normally distributed, the Student $t$ test. If comparing a normally distributed distribution with a nonnormal distribution, nonparametric tests were applied. Statistical significance stars were allocated according to the convention; ns ($P > 0.05$), * ($P \leq 0.05$), ** ($P \leq 0.01$), *** ($P \leq 0.001$), and **** ($P \leq 0.0001$).

## Results

### Principles and construction of IVEN

Three fundamental embryo-scale architectural changes occur during the preimplantation period: compaction, cavitation, and hatching from the zona pellucida (Fig 1A). Such architectural changes are largely described qualitatively in the literature, and a tool to assess changes in cell neighbourhood during such events is not yet widely available. Some existing studies have attempted to describe embryo architecture quantitatively, using methods such as the neighbourhood analysis of ICM cells as presented in [21] and manual assessments of epithelial TE cell layer neighbourhoods in [22]. However, to our knowledge, no tool or system published so far provides a user-friendly, automatic assessment of the position and neighbourhood of every single cell within the 3D environment of the preimplantation embryo. Therefore, we designed IVEN, which is able to generate embryo schematics, classify cell position as inside or outside, and subsequently calculate the number of neighbours for each cell in 3 dimensions (Fig 1B and 1C, S1 Fig). All code and example data are available at https://github.com/jessforsyth/IVEN-code.

A key objective during the development of IVEN was to ensure its use across different developmental stages and model systems. Therefore, IVEN was designed as a relatively simple tool and only requires the 3D coordinates of cell centres (or nuclei centres) in order to perform cell classification and the calculation of the number of neighbours (Fig 1C, S1 Fig). Cell centres can be extracted using existing protocols such as IMARIS spot detection, MINS [13], ImageJ segmentation tools, or Nessys (use of nuclear envelope) [14], and then imported into IVEN. To aid visualisation and analysis of the data, we incorporated the import of channel intensities and cell identification numbers from segmented confocal images so that protein abundance could also be visualised in the IVEN interface and specific cells analysed in conjunction with standard image analysis techniques.

Existing methods to distinguish outside (external) and inside (internal) cells in the murine embryo use sophisticated ellipsoid fitting techniques [13]. However, we found that the fitting of the ellipsoid can prove erroneous in some embryos due to deformation, or more irregular shapes of later stage embryos, as well as partial hatching from the zona pellucida. In addition, while the convex hull approach we employ within IVEN is relatively simple, some ellipsoid fitting techniques such as that included in [13] often rely on additional fitting techniques such as random sample consensus methods. The convex hull is defined as the minimum set of points that when joined together enclose all points within the dataset [23,24] and so provides a good initial description of the external cells (in our case, TE). The convex hull has also previously been successfully used to identify regions of the lumen/cavity within the preimplantation embryo [25]. Since embryos can vary in their overall morphology, we included a manual correction step, so that users can manually reallocate cells to the inside or outside groups where necessary, using a graphical user interface (S1 Fig).

To quantitatively describe embryo architecture, we chose to use the number of neighbours as a measure/description of the cell environment as in [21]. To calculate the number of neighbours for each cell, we employed a corrected 3D DT [16]. The standard DT describes a network between cells that is constructed from a set of tetrahedrons according to specific geometrical rules [16,26]. The DT poses as a mathematical description of "neighbourhood" for a given set of 3D points, thus removing subjectivity as to what criteria a cell must meet in order to be described as a neighbour (for example, area of cell-to-cell contact). However, the DT does not account for potential cavities present within samples (S2A and S2B Fig). Therefore, to preclude erroneous classification of cells positioned on opposite walls of cavities as neighbours, we modified the IVEN neighbourhood pipeline. We imposed a distance-based

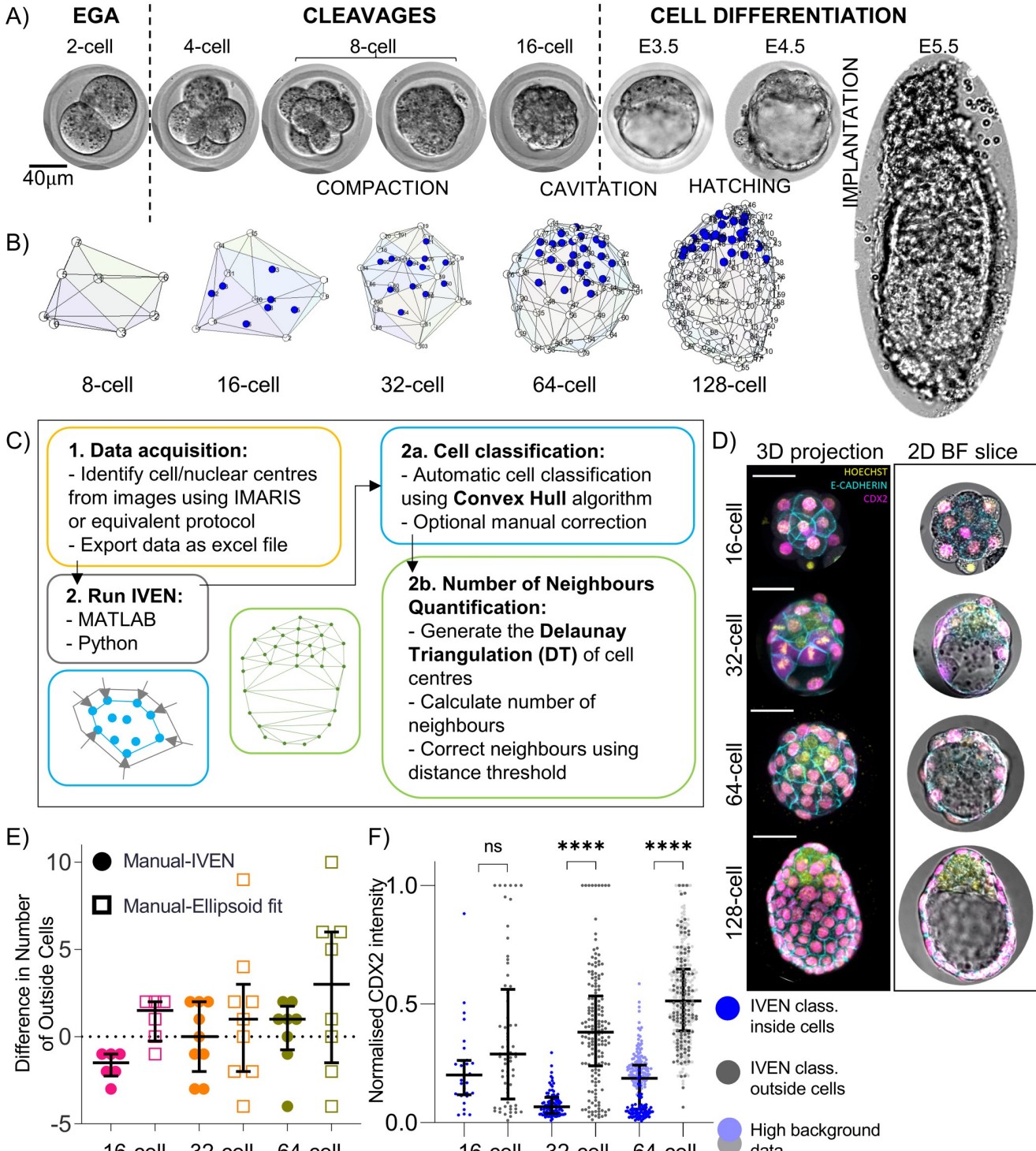

**Fig 1. IVEN development and implementation in preimplantation stages of the murine embryo.** (A) Brightfield images of the preimplantation and peri-implantation periods of the murine embryo. Scale bar, 40 μm. (B) Embryo reconstructions of murine embryos at the 8-, 16-, 32-, 64-, and 128-cell stages generated within the IVEN pipeline. Outside cells are shown in white, and inside cells are shown in blue. Numbers labelled are cell IDs. (C) IVEN pipeline; data preparation (yellow box), selection of platform (grey box), cell classification (blue box), and neighbourhood quantification (green box). Blue inset shows an example 2D convex hull around 2D coordinates. Green inset shows an example DT of in silico 2D data points approximating a cross section through a blastocyst. (D) E-cadherin and CDX2 expression between the 16-cell stage and 128-cell stage. Restriction of CDX2 to outside (TE) cells by the 32-cell stage. Scale bar, 40 μm. (E) Difference between number of outside cells as classified using IVEN (convex hull and manual reclassification where necessary, circles), ellipsoid fitting (squares), and manual identification of outside cells based on CDX2 expression and positional information per embryo.

Each marker is representative of the difference in the number of outside cells for one embryo. The dotted line indicates a difference of zero. (F) Comparison of outer cell versus inside cell CDX2 expression as classified using the IVEN pipeline. Dark blue and dark grey points indicative of low background data. Pale blue and pale grey points indicative of high background data analysed at the 64-cell stage. Data underlying this figure can be found on the public GitHub repository https://github.com/jessforsyth/forsyth-et-al-2021. DT, Delaunay triangulation; EGA, embryonic genome activation; IVEN, Internal Versus External Neighbourhood; TE, trophectoderm.

threshold on neighbours, where the threshold represents a maximum allowed distance between neighbours, in the units of the imported data (pixels or μm). If the distance between 2 neighbours, identified using the DT, was greater than the threshold, the cells were not considered as neighbours (S2B Fig).

IVEN was developed to offer several neighbour thresholding options, including user-defined distances as well as thresholds inferred from the dataset. For our analyses, thresholds were calculated automatically by IVEN using the cell centre coordinates on an embryo-specific basis. Euclidean distances between all original identified neighbours within an embryo were calculated and plotted as a distance distribution (S2C and S2D Fig). Then, the threshold was set to equal $P_{75} + (k \times \text{IQR})$, where $P_{75}$ is the 75th percentile of the neighbour distance distribution and IQR is the interquartile range of the distribution. Neighbour distance thresholds were calculated separately for inside and outside cells from the 32-cell stage onwards. We then chose $k$ to equal 0.5 after tuning through comparison to manual neighbourhood assessments (S2E Fig).

The pipeline was developed in both Python and MATLAB to encourage users from different backgrounds to interact with the code and offer different running options. The MATLAB version of the pipeline was more graphically interactive, thus meaning there is less requirement for any coding knowledge, though this also resulted in the MATLAB version being more graphically intensive. As both systems are largely interactive, we developed runtime tests where the user interfaces were generated but automatically closed (S2F Fig). It was evident that both versions run in appropriate times for moderate numbers of cells that easily covered the expected number of cells in embryos during the murine preimplantation period.

## Successful classification of TE versus ICM cells using IVEN

To test IVEN's performance in classifying inside and outside cells, we analysed mouse embryos collected at the 16-, 32-, and 64-cell stages. Embryos were stained for E-cadherin and CDX2 to visualise cell–cell contacts and aid manual identification of outside cells. To enable visualisation of chromatin, embryos were also stained with Hoechst, which was used to identify nuclear centres for input into the IVEN pipeline.

CDX2 was expressed in most cells at the 16-cell stage and marked the TE exclusively by the 32-cell stage as previously reported (Fig 1D) [27–29]. We identified CDX2 positive cells manually using the IMARIS 3D visualisation for all the investigated stages (Fig 1D). At the 16-cell stage, cells were classified using both CDX2 expression and manual positional scoring as CDX2 expression at this stage was not completely restricted to outside cells. We compared a simple ellipsoid fitting technique (Yury, Ellipsoid fit (https://www.mathworks.com/matlabcentral/fileexchange/24693-ellipsoid-fit), MATLAB Central File Exchange, retrieved May 24, 2021), with the convex hull algorithm to verify our selection of cell classification method within the IVEN pipeline. We used the number of outside cells counted manually using the CDX2 and positional scoring as the ground truth number of outside cells. The number of outside (TE) cells counted using IVEN was only slightly different at the 16-, 32-, and 64-cell stages (Fig 1E). However, when we compared the difference in the number of outside

cells identified using the ellipsoid fitting technique, it was evident that there were typically larger differences in the number of outside cells compared to the convex hull (IVEN) technique (Fig 1E). We were therefore satisfied that the convex hull algorithm was the optimum method to classify cells within the IVEN pipeline.

To further validate the convex hull approach, we compared the normalised intensity of CDX2 fluorescence between cells classified by IVEN using the convex hull as inside and outside. All stages except the 16-cell stage showed significantly higher expression of CDX2 in cells classified as outside cells, indicating the successful classification of TE cells (Fig 1F). Encouragingly, at the 64-cell stage, some data had a high background intensity but still displayed higher CDX2 expression in the outside cell population (Fig 2F, S3 Fig). The nonsignificant difference in CDX2 expression at the 16-cell stage was expected, as CDX2 expression was not yet restricted to outer cells at this stage, resulting in similar expression profiles in the inside and outside cell groups [27,28].

We demonstrate that the convex hull is an accurate, robust, and simple approach to classify cell populations within the preimplantation embryo. The convex hull algorithm is a flexible method of cell classification that is able to accommodate non-ellipsoid morphologies as well as more regular convex shapes. In conjunction with this, users can use inbuilt user interfaces to further refine cell classification manually within IVEN. This simple approach helps classify populations of cells that are potentially in contact with the external environment and reduces the subjectivity of manual cell classification alone.

## TE cells typically exhibit fewer neighbours than ICM cells

As we were able to successfully classify inside versus outside cells using IVEN, we set out to describe the changes in neighbourhood of inside and outside cells within the preimplantation embryo. Embryos were collected at the 8-, 16-, 32-, 64-, and 128-cell stages (in addition to those previously analysed for outside–inside classification) (Fig 2A) and were stained using Hoechst dye to enable visualisation of the chromatin. After imaging, we used the spot detector in IMARIS to identify nuclei centres to approximate cell centre positions within the embryo. Embryos chosen for analysis had within 10% of cells of the desired developmental stage; for example, "32-cell stage" embryos selected had 32 ± 3 cells. All analyses were run using the MATLAB version of IVEN, and example embryo reconstructions for each stage including cell classification were generated (Fig 1B, S1–S5 Videos).

Embryos with 8-cells were manually checked for signs of compaction; 7 out of 48 embryos were found to be compacted. The numbers of neighbours of cells and calculated neighbour distance thresholds were compared between compacted and noncompacted groups and were shown to have no statistically significant differences (S4A and S4B Fig). We therefore grouped all 8-cell stage embryos together. The similarity between compacted and noncompacted morulae was consistent with the notion that the IVEN pipeline uses the nuclear centres, and these centres are unlikely to move significantly during compaction due to the high cell to nucleus size ratio.

At all stages where an inside population of cells existed, outside cells had on average fewer neighbours than inside cells (Fig 2B). This was consistent with expectation that outside cells would have a reduced number of neighbours due to a proportion of their cell surface being in contact with the external environment, thus resulting in fewer neighbours. Our analyses further supported the ability of the IVEN pipeline to classify inside and outside cell populations correctly, leading to the quantification of distinct neighbourhoods, without the need for lineage specific protein markers.

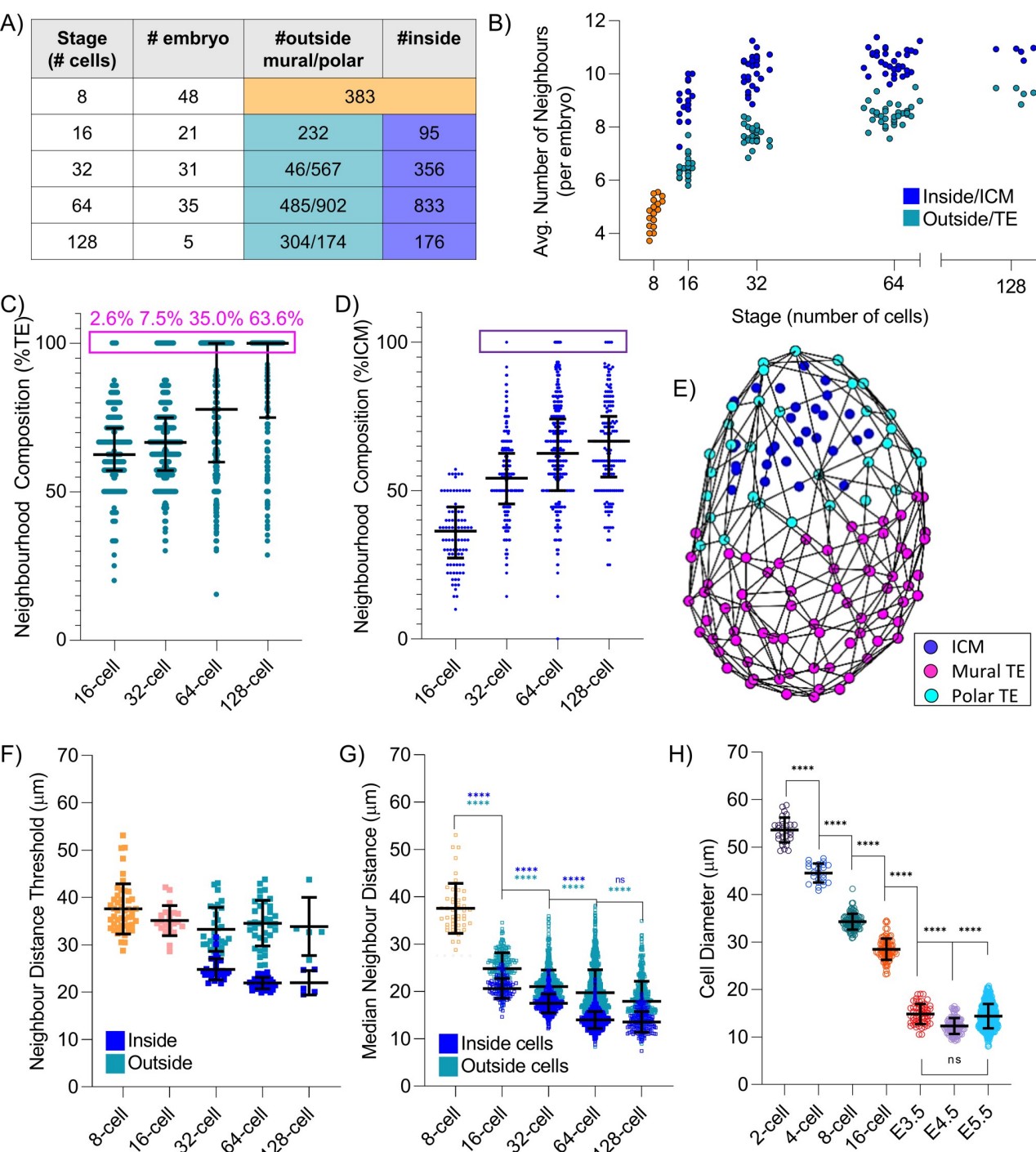

**Fig 2. Assessment of the neighbourhood of preimplantation embryos between the 8-cell stage and 128-cell stage using the IVEN pipeline.** (A) Number of analysed embryos and cells per developmental stage that was assessed using IVEN. (B) Average number of neighbours of outside and inside cells per embryo. Outside cells shown to typically have fewer neighbours than inside cells. (C) Neighbourhood composition of outside cells presented using the percentage of neighbours that were also outside cells. Outside cells shown to exhibit a subpopulation of cells with 100% of their neighbours also outside cells (pink box). (D) Neighbourhood composition analysis of inside cells, as described by the percentage of neighbours that were also inside cells. General increase in the percentage of inside cell neighbours that are also inside cells. Inside cells shown to exhibit a subpopulation of cells with 100% of their neighbours also inside cells (purple box). (E) Embryo schematic of a 128-cell stage embryo with mural versus polar TE cells marked, as classified by IVEN. (F) IVEN calculated thresholds for each developmental stage. Each marker is representative of an embryo threshold. (G) Median distance to neighbours for inside and outside cells. (H) Change in average cell diameter with developmental stage. Cell diameter decreases significantly at all stages until E4.5 where cells then appear to increase in size at the E5.5 stage. Cells appear of similar diameters at E3.5 and E5.5. Data underlying this figure can be found on the public GitHub repository https://github.com/jessforsyth/forsyth-et-al-2021. ICM, inner cell mass; IVEN, Internal Versus External Neighbourhood; TE, trophectoderm.

## Mural and polar TE distinguished by neighbourhood composition analyses

Our ability to describe neighbourhood and cell position using the IVEN pipeline led us to extend our analysis further by quantifying the neighbourhood composition of inside and outside cells. We calculated the proportion of neighbours of outside cells that were also outside cells. We identified populations of outside cells at blastocyst stages that had neighbourhoods exclusively made of other outside cells (Fig 2C). By definition, mural TE cells have exclusively TE neighbours due to the presence of the cavity, whereas polar TE are likely to have neighbours that are both TE and ICM cells. We mapped the cell allocation (mural versus polar) onto the embryo schematics and found that the subpopulation identified was indeed the mural TE (Fig 2E, S3–S5 Videos). At the 16-cell stage, we did also detect some outer cells surrounded only by outer cells. The existence of such cells support the notion that there are very few inside cells detected at the early 16-cell stage as observed in [30], ultimately leading to some outside cells only being in contact with other outside cells (S5A and S5B Fig).

We were able to clearly show the emergence of the mural TE cells through neighbourhood composition analyses and calculate the proportion of the TE that formed the mural population (Fig 2C and 2E). The average percentage of mural TE cells increased from 7.5% to 63.6% between the 32- and 128-cell stages, supporting the observation that the mural TE population increases during cavity expansion (Fig 2C, S5C Fig) [31,32]. The number of mural TE identified at the 32-cell stage was lower than expected, and, upon closer analysis, it was found that not all embryos had fully developed cavities, leading to some embryos having zero mural TE cells (S5D–S5H Fig). When we compared the total number of cells in 32-cell stage embryos with the number of identified mural TE, we noticed no correlation (S5D Fig). This is consistent with previous observations that in early stages of cavity expansion, there is no dependence of cavity expansion on the total cell number [11].

By studying the outside cell neighbourhood composition, we were able to interrogate the dynamic of mural and polar TE formation and confirm that initial cavity formation and expansion is not directly related to the cell number in the mammalian embryo.

## Deep ICM cell population increases during blastocyst development

Continuing this notion of neighbourhood composition analysis, we performed a neighbourhood composition analysis for inside cells and identified subpopulations of inside cells that had 100% of their neighbours that were also inside cells (Fig 2D). We observed that the average percentage of neighbours of inside cells that are also inside cells increased between the 16- and 128-cell stage, up to an average inside cell neighbourhood composition of approximately 65% inside cells. These data suggested that the number of deep ICM cells, those entirely surrounded by other inside cells, could potentially increase during development, and that more inside cells are either isolated from or in minimal contact with outside cells.

The subpopulation of inside cells surrounded entirely by other inside cells may be subject to differential signalling regimes leading to initiation of cell specification and organisation and could play an important role in driving or facilitating ICM specification into Epi or PrE cells.

## Distance between cell neighbours decreases with developmental stage

During the analysis of embryos using IVEN, maximum neighbour distance thresholds were calculated for inside and outside cell groups, respectively. According to our expectation, thresholds for inside and outside cells appeared to decrease with developmental stage in a manner that mirrored an expected decrease in cell diameter due to cell cleavages during the preimplantation period (Fig 2F). We hypothesised that this reduction in threshold could be attributed to the decreasing cell size during the preimplantation period. To study this further,

we calculated the median distance between each cell and its neighbours using the IVEN pipeline. We found that typically, inside cells had smaller distances between neighbours than outside cells (Fig 2G). This could suggest that cells are more closely packed within the ICM or further demonstrate the effect of the stretching of the TE during cavity expansion. We also noticed that the median distance between a cell and its neighbours decreased with developmental stage, similar to the manner in which the maximum neighbour distance threshold decreased. The distance between neighbours was significantly smaller with each developmental stage except between inside cells at the 64- and 128-cell stages, though the median distance to neighbours was reduced at the 128-cell stage (Fig 2G). This could potentially suggest that cells reach a minimum diameter around the time of implantation, resulting in an observed decrease in distance between neighbours until this point.

## Cell diameter decreases with developmental stage until implantation

To better understand the relation between the distances between cells and their neighbours and the changes in cell size in development, we decided to quantitatively describe changes in cell diameter between the 2-cell stage and the blastocyst stage (E3.5). As the exact developmental stage at which cells stop decreasing in size and first initiate a growth phase within the cell cycle in mammalian development is not clearly defined, we extended our analysis to the periimplantation period and collected implanting embryos (E4.5) as well as early postimplantation embryos (E5.5) (S6A Fig).

In order to obtain cell diameter measurements unaffected by cell packing within the embryo, embryos were disaggregated prior to cell diameter measurements. Earlier embryos (up to and including E3.5) were disaggregated using calcium-free M2, and later-stage embryos were treated with trypsin (E4.5 and E5.5). To visualise the cell membrane, cells were stained using the FM4-64FX dye, similar to the dye used in [19] (S6A Fig). After disaggregation, cells rounded up (S6A and S6D Fig). Therefore, we assumed the diameter across the z-slice with the maximum cross-sectional area of the cell to be representative of cell size. We measured 2 perpendicular axes of the disaggregated cells and averaged these measurements to provide a measure of the cell diameter (S6B Fig).

Between the 2-cell stage and E4.5, cells continued to decrease in diameter (Fig 2H) similar to the observed decrease in distance between cells and their neighbours (Fig 2G). However, between E4.5 and E5.5, we observed an apparent increase in average cell size, such that the E5.5 blastomeres appeared to be of similar sizes to cells in E3.5 embryos. To check that we did not include measurements from embryos that were not at the desired developmental stage, we compared the cell diameter measurements on an embryo-to-embryo basis (S6E–S6K Fig). No data points from a single embryo were obvious outliers; thus, we were able to conclude that the embryos sampled were of the desired stages.

Our observations, therefore, strongly suggest that the transition from cleavage division to division more akin to somatic cell cycle coincides with the time of implantation. This mirrors the observed decrease in distances between neighbours calculated by IVEN and suggests that the decrease in distances between neighbours is largely due to the decrease in cell size over the preimplantation period.

## Distinct cell neighbourhoods change present within the preimplantation period

To understand better how changes in overall embryo architecture relate to TE (mural and polar) and ICM cell environment, we investigated cell neighbourhood dynamics between the 8-cell stage and the time of implantation. We calculated the number of neighbours of cells

over this developmental period by using IVEN and applying the IVEN-defined classifications of cells (mural, polar, or ICM) (Fig 3A). Cells from 8-cell stage embryos had a median of 5 neighbours, which increased to a median of 9 neighbours for inside cells from 16-cell stage inside cells (Fig 3A). The median number of neighbours of inside cells was shown to increase very slightly from 10 neighbours at the 32-cell stage up to 11 neighbours at the 128-cell stage (Fig 3A). We noticed a general increase in the number of neighbours for all cell groups, potentially indicative of changes in cell packing/organisation.

To further extend our description of cell environment for each of the cell groups, we generated neighbourhood frequency distribution plots (Fig 3B–3F). These analyses allowed for the assessment of the proportions of cells with a given number of neighbours with each cell group (per stage) and provided a comprehensive understanding of the changes in cell neighbourhood during development. We found that 33.4% of cells from 8-cell stage embryos had 4 neighbours but that some cells could have up to 7 neighbours (Fig 3B). At the 16-cell stage, outside cells and inside cells were shown to have significantly different neighbourhood distributions, and outside cells were shown to typically have fewer neighbours than inside cells as alluded to in previous analyses (Fig 3C).

In blastocyst stage embryos, it was evident that the majority of inside cells had between 9 and 12 neighbours. The neighbourhood frequency distribution analyses revealed that 72.2%, 67.9%, and 64.8% of inside cells from the 32-, 64-, and 128-cell stages had numbers of neighbours within this range. The distributions of polar and mural TE neighbourhoods differed from the inside cell distributions as they typically appeared more peaked around singular modal value of numbers of neighbours rather than a broadened peak as observed for inside cells (Fig 3D–3F). Mural TE cells typically had 5 (28.3%), 6 (28.5%), and 8 (23.7%) neighbours at the 32-, 64-, and 128-cell stages, whereas polar TE cells typically had 8 (27.0%), 9 (22.3%), and 9 or 11 (18.4%) neighbours at the 3 blastocyst stages, respectively. As mural TE can be treated as a single layer epithelium, our data seem to suggest that the mural TE achieves the previously described optimal hexagonal cell packing regime around the 64-cell stage and that this packing regime changes around the time of implantation (128-cell stage). Our data also suggest that mural and polar TE cells have a more confined or regular packing than inside cells due to the presence of a more peaked neighbourhood frequency distribution.

Inside cells, mural TE and polar TE cells were mostly shown to have significantly different neighbourhood distributions at the 32-, 64-, and 128-cell stages (Fig 3D–3F). However, at the 128-cell stage, the polar TE and inside cell neighbourhood distributions were shown to be similar and no longer statistically significant from one another. This could be a feature of the embryo being ready for implantation or simply the effect of the ICM, resulting in more similar neighbourhood frequency distributions.

## Modulation of FGF pathway alters spatial relations between ICM cells

IVEN allows for fatefully and efficient analysis of cell position and neighbourhood within the embryo, alongside descriptions of changes at the cellular level. We therefore decided to test whether IVEN could be used to detect changes in cell distribution in embryos where the morphology of the ICM was perturbed. Treatment of rabbit and mouse embryos with FGF4 has been shown to cause the majority of ICM cells to adopt PrE fate and induce morphological changes in the embryo, generating "flattened" and more dispersed ICMs [33,34]. How this change in cell fate allocation and global ICM geometry translates to cell packing and the number of neighbours of cells remains unclear. We therefore sought to use IVEN to describe these changes in FGF4-treated murine embryo and identify any observed quantitative differences in cell neighbourhood and distribution across treatment regimes.

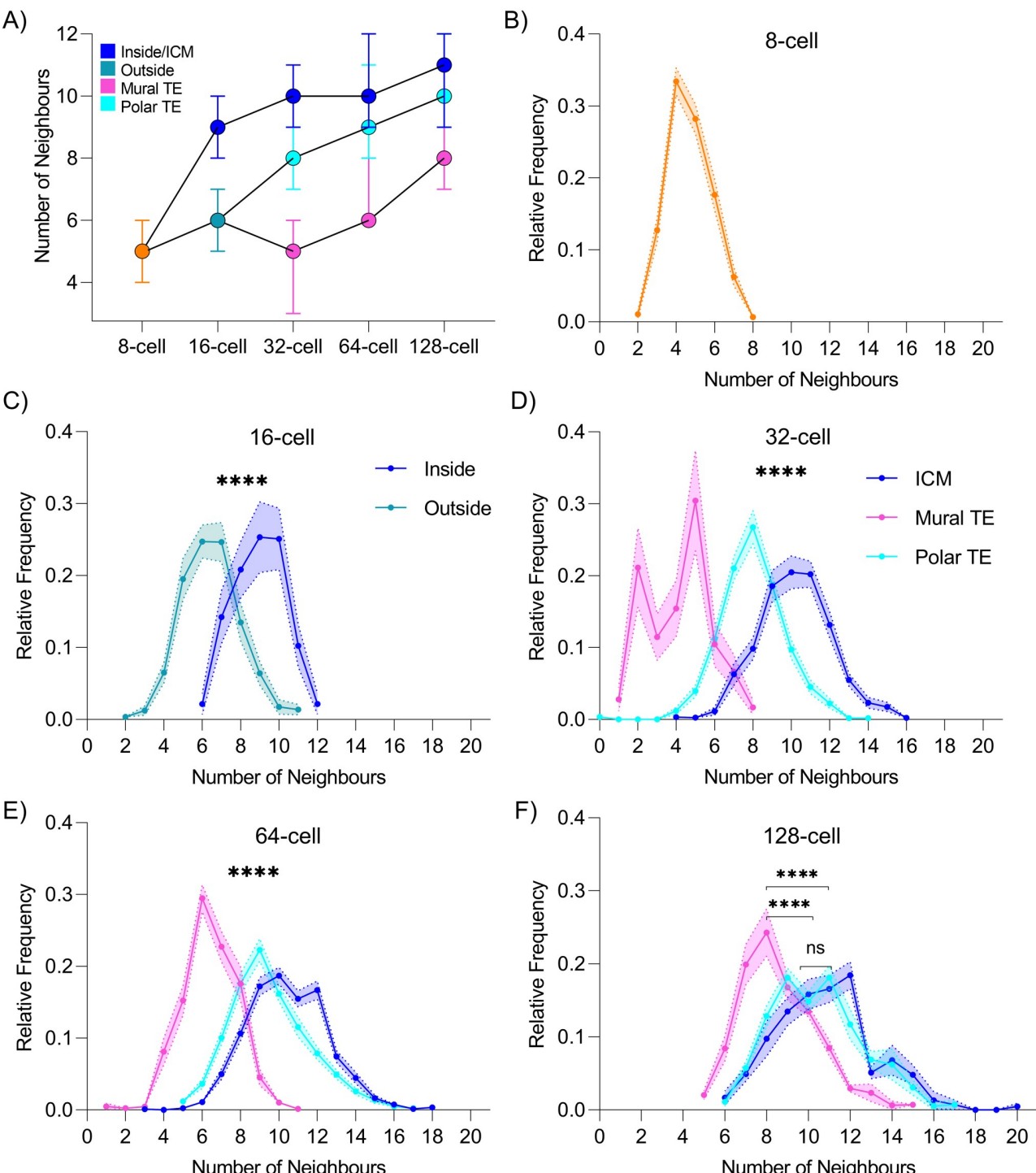

**Fig 3. Neighbourhood analysis and frequency distributions at each developmental stage from the 8-cell stage up to the 128-cell stage.** (A) The overall change in neighbourhood for inside and outside (mural or polar TE) cells between the 8 and 128-cell stages. (B–F) Average frequency distribution with shaded regions indicative of the standard error of the mean at the 8- (no inside or outside populations categorised), 16-, 32-, 64-, and 128-cell stages, respectively. Relative frequency refers to the normalised frequency (or probability) for each number of neighbours. Data underlying this figure can be found on the public GitHub repository https://github.com/jessforsyth/forsyth-et-al-2021. ICM, inner cell mass; TE, trophectoderm.

Embryos were collected at the 8-cell stage and subject to culture in KSOM (control) ($n = 16$ embryos) or KSOM+FGF4 and heparin sulphate ($n = 15$ embryos) to produce embryos with ICMs predominantly composed of PrE cells, as described previously in [20]. Embryos were cultured for approximately 48 hours until they reached E4.5 and were fully hatched from the zona pellucida. A subset of embryos were then fixed and stained for NANOG and GATA4 as markers for the Epi and PrE lineages within the ICM respectively (Fig 4A).

Qualitatively, changes to the ICM morphology were evident in FGF4-treated embryos with the already described "flattening" of the ICM against the TE and ICMs nearly entirely composed of GATA4+ cells (Fig 4A). The number of cells within the ICM (as classified by IVEN) was shown not to change significantly between the 2 culture conditions (Fig 4B). We used IVEN to automatically generate neighbour distance thresholds for the inside cells, classify cells as inside or outside, and calculate the numbers of neighbours of the inside cells. When comparing the median distance between inside cells and their neighbours between the 2 treatment conditions, we noticed that cells and their neighbours were typically further apart in FGF4-treated embryos (Fig 4C). This suggests that inside cells in FGF4-treated embryos are more dispersed and on average further from each other.

To further investigate potential differences in ICM neighbourhoods between the culture conditions, we studied the neighbourhood composition of inside cells. We calculated the median percentage of inside cell neighbours that were also inside cells for each embryo and compared KSOM- and FGF4-treated embryos (Fig 4D). The KSOM-cultured group was shown to be significantly different from the FGF4-treated group. Inside cells from KSOM-cultured embryos were shown to exhibit neighbourhoods with higher numbers of neighbours that are also inside cells. We also show the neighbourhood composition of every inside cell (not averaged) (Fig 4E). We still found that KSOM inside cell neighbour compositions were significantly different from FGF4 treated inside cell compositions. Interestingly, we identified a subpopulation of cells that had 100% of their neighbours also inside cells (Fig 4E), like the population previously found in wild-type, freshly flushed embryos (Fig 2D). This population was greatest in the KSOM culture condition, with 5.6% of inside cells analysed exhibiting this feature, and only 0.3% (one cell from the entire group), of inside cells analysed from FGF4-treated embryos. This feature of the inside cell environment was clearly affected during FGF4 treatment and could reflect the flattened ICM morphology, which makes cells with 100% of their neighbours also being inside cells far less likely to occur.

When comparing the neighbourhood frequency distributions, we noticed changes in the proportions of cells with 9 or 10 neighbours (Fig 4F). Although the distributions occupy similar peak values, somewhere between 8 and 12 neighbours, it was evident that the distribution for the FGF4 treatment appeared more peaked around 9 and 10 neighbours, whereas the distribution for the KSOM group appeared broader. This peaked neighbourhood distribution of inside cells in FGF4-treated embryos could potentially be linked to formation of a "double layer" epithelium that consists of TE and PrE, mirroring our previous results whereby TE (mural and polar) groups displayed more peaked neighbour distributions in comparison to the ICM groups. In summary, our data demonstrate that FGF4-induced changes in ICM cell identity and ICM geometry are accompanied by subtle changes in cell neighbourhood and more evident changes to neighbourhood composition, as well as inside cell dispersion within the embryo as described by the median distance between neighbours.

## Discussion

In the early mammalian embryo, subsequent rounds of cell divisions lead to constant remodelling of cell microenvironment. This, in conjunction with profound changes in embryo

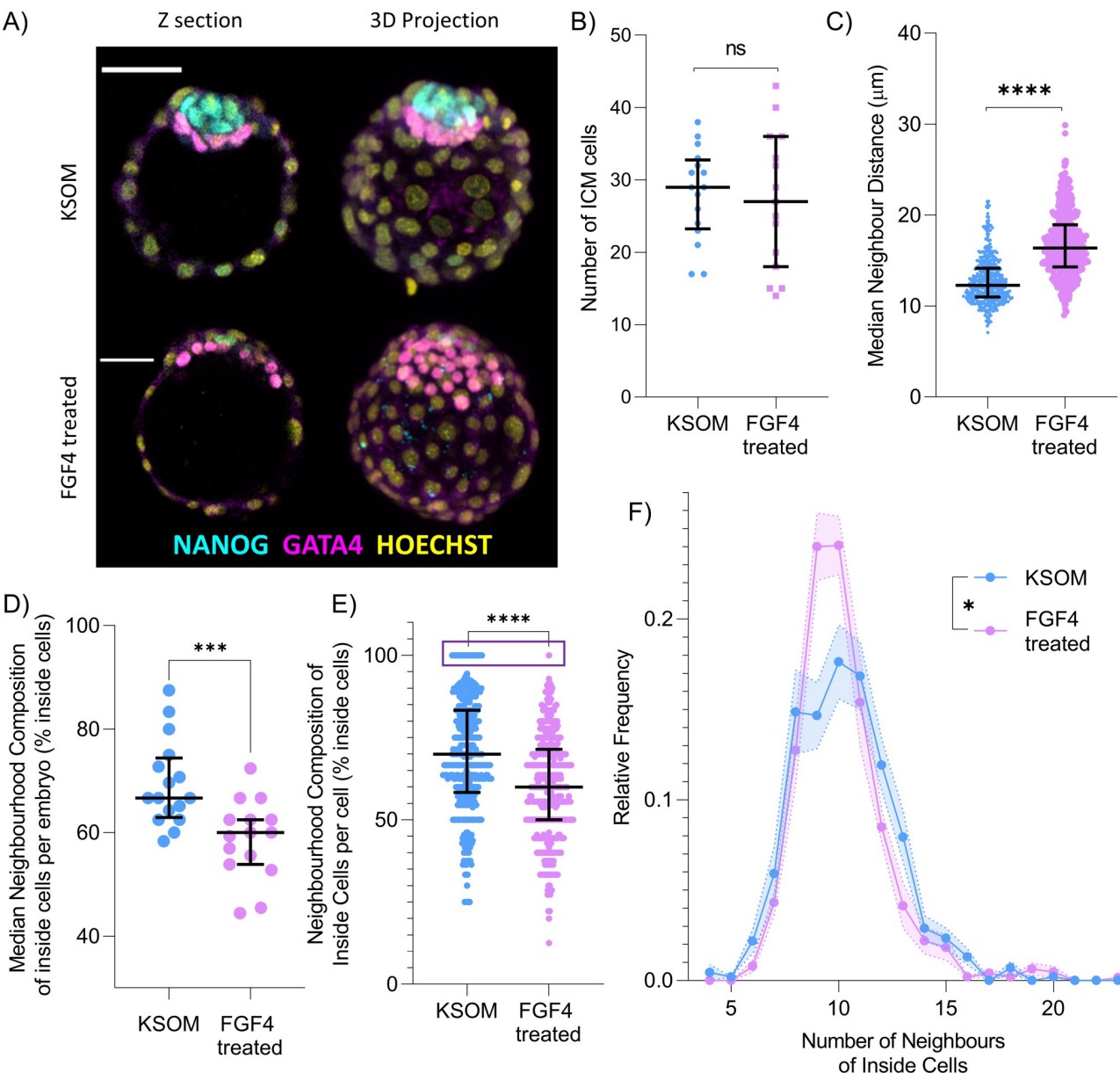

**Fig 4. Comparison and quantification of morphological changes between embryos cultured in KSOM (control) and in KSOM+FGF4.** (A) Expression of NANOG (Epi marker) and GATA4 (PrE marker) in the control group, compared with predominant expression of GATA4 in embryos cultured with FGF4. A more extended, flattened ICM is evident in embryos cultured with FGF4 when compared to embryos cultured in KSOM. Scale bar, 40 µm. (B) Comparison of the number of ICM cells in KSOM-cultured embryos versus FGF4-treated embryos as classified by IVEN. (C) Comparison of the median neighbour distance between inside cells and their neighbours within the 2 treatment regimes. (D) The median value of the neighbourhood composition for inside cells per embryo, given as the percentage of neighbours that are also inside cells. (E) Neighbourhood composition of all inside cells analysed showing the reduction in the number of inside cells with 100% of neighbours also inside cells in the FGF4-treated group (purple box). (F) Average neighbourhood frequency distribution of inside cells with shaded regions indicative of the SEM, comparing culture conditions. Higher proportion of cells in FGF4 treatment conditions have lower numbers of neighbours when compared to KSOM-cultured embryos. Relative frequency refers to the normalised frequency (or probability) for each number of neighbours. Data underlying this figure can be found on the public GitHub repository https://github.com/jessforsyth/forsyth-et-al-2021. Epi, Epiblast; ICM, inner cell mass; IVEN, Internal Versus External Neighbourhood; PrE, primitive endoderm.

architecture during preimplantation development, provides a powerful interactive framework that influences cell signalling and cell fate. So far, our understanding of processes driving pre-implantation mammalian development was built largely around the action of specific transcription factors and signalling pathways using a variety of biochemical and genetic approaches, with some attempts to qualitatively describe morphological changes such as lumen expansion during blastocyst formation [8,25,35]. Taking into consideration all previously published data, it is evident that physical forces and the cell microenvironment play a vital role in development [21,35–37]. Recently, it has been proposed that during compaction, the cellular and global changes within the embryo help specify the inside cell population (future ICM) due to internalisation of apolar cells following asymmetric cell divisions [7,30,38]. Similarly, the distribution of cells within the ICM and the formation of specific cell microenvironments have been linked to specification of PrE and Epi lineage in the developing mouse embryo [21]. In addition, differences in cell shape and size arising in preimplantation development have been suggested to be involved in the lineage specification process [38,39].

Despite the clear importance of the spatial and geometrical cues in development, the effects of changes in cell organisation and embryo geometry are still not explored fully with regard to mammalian development. The main reason is the lack of readily accessible, simple to use software that allows the quantification of spatial cell distributions within multicellular systems. As a result, most of the existing analyses of morphology have been performed manually, which precludes large-scale analysis. Several elegant studies attempted to address the issue of cell packing and cell neighbourhood changes in dynamic developmental systems [14,21]. These approaches used local cell density assessments to analyse cell culture architectures [14] and Delaunay cell graphs to assess cell neighbourhoods in subsections of preimplantation embryos [21]. These studies gave some of the first insights into the architectural features of cell culture systems and the preimplantation embryo, respectively. However, in the case of more complex structures, like the entire mammalian blastocyst, the previously suggested approaches cannot be readily adapted. As we were unable to find user-friendly, freely available programs to assess 3D cell position and the neighbourhood of all cells within embryos, we developed the pipeline, IVEN. We designed IVEN to be used by biomedical scientists with minimal training in programming and so include basic, but comprehensive, user interfaces within the pipeline. IVEN was developed in MATLAB and Python to provide users with the choice of platform and allow for potential integration into existing data analysis pipelines. We were also aware that though most academic institutions provide access to MATLAB, this can sometimes be restricted to postgraduate researchers and staff. We therefore provide a Python version of IVEN as the Python software is freely available to all users. The MATLAB version largely provides a user-interactive platform with little/no coding knowledge necessary, whereas the Python version is more stripped back, with interfaces incorporated only for cell classification checking and file selection. However, this version, due to its stripped-back nature, can be faster and able to process larger datasets than the current MATLAB version due to a lower graphical demand.

IVEN was designed as a tool to mathematically describe individual cell environments and classify cells as either external or internal. To classify cell position, we implemented a convex hull approach with subsequent user scrutiny using an interactive user interface. We believe the convex hull to be superior when compared to other methods of geometrical cell classification, such as simplistic ellipsoid fitting techniques, as the convex hull can easily adapt to embryos/tissues where some deformation is present [23,24]. With further development, and more sophisticated fitting algorithms such as those used in [13], the ellipsoid fitting technique can prove to be an accurate method of cell classification, but for the purpose of IVEN and increasing the flexibility of the pipeline, we opted to use the convex hull algorithm. The interactive user interface was incorporated to accommodate for greater levels of deformation that can

sometimes occur in larger, more fragile embryos and to make sure that the program can be used for a variety of purposes and tissues/cell aggregates. To ensure that IVEN was able to robustly categorise cells as inside or outside using the convex hull, we performed manual cell classification based on the presence (TE cells) or absence (ICM cells) of the TE marker CDX2 and confirmed that IVEN correctly classifies cells to the "inside" or "outside" categories for the various stages of the preimplantation mouse embryo. The convex hull was shown to be accurate in classifying outside cells especially at the 32- and 64-cell stages, where few cells needed to be manually reclassified. One could therefore surmise that the TE in the mouse embryo can be approximated as the convex hull of the system. Such mathematical descriptions of the TE could influence mathematical, geometrical modelling of the preimplantation embryo, and, further, our understanding of energy constraints within 3D epithelial layers.

The Voronoi formalism and its relative, the DT, have been useful tools for the analysis of both 2D and 3D cell neighbourhoods and mechanisms underlying tissue organisation [40–42]. Therefore, in order to assess the neighbourhood of cells, we decided to use the DT, an approach analogous to that employed previously in [21,43]. The DT does not take into consideration the boundaries of cells unlike approaches using trijunctions or regions of contact between cells [22,37]. These alternative approaches are effective in 2D systems but increasingly difficult to apply for 3D confocal images due to the attenuation of signal and limiting z resolution, especially when using live samples and time-lapse technologies. These existing approaches also rely on user definitions of neighbours, leading to subjectivity during analysis. By using the DT and a more mathematically defined idea of "neighbourhood," based on a unique DT map generated from nuclear centres, we believe that we avoid the ambiguities that arise from manual assessment of cell environment, and, as a result, can give a more complete, quantitative description of cell environment.

In general, using the DT presents one problem when embryos, tissues, or organoids with cavities and lumens are to be analysed. In the case of preimplantation mammalian embryos, when the DT is constructed for blastocyst stage embryos, some connections between cell centres span across the blastocyst cavity, resulting in what we referred to as "untrue" neighbours. As we wanted to include mural TE cells (those that surround the cavity) in our analyses, we implemented a neighbour distance threshold to ensure that cells across the cavity were not considered neighbours erroneously. The previous approach in [21] also applied a distance threshold but also removed any TE cells that had no contact with inside cells, which, although appropriate for their study, would not allow us to fully describe changes in the neighbourhood for all cell types. Cells positioned across the cavity from each other typically had larger neighbour distances than true neighbours and so we developed a method to automatically calculate the most appropriate neighbour distance threshold to apply. This approach gave a greater degree of flexibility to the IVEN pipeline as the threshold could be calculated on a sample-to-sample basis. The thresholding approach was further extended by incorporating cell type–specific thresholds, i.e., generating separate thresholds for inside and outside cells. By calculating separate thresholds for inside and outside cells, we were able to account for stretched TE cells that typically had larger distances between neighbours and ensure that appropriate thresholding was performed for both inside and outside cells. Further to this, using the data-defined thresholding approach allowed for accommodation of differences in coordinate base that could occur during imaging, potential deformation of embryos, difference in cell sizes, and different sizes of the cavity could be easily accounted for. During calculation of the thresholds at the 32-cell stage, we noticed that the distance between mural TE neighbours and cavity-bounding ICM cells sometimes appeared similar to the diameter of the cavity during early cavitation. This led to some difficulties in correction of the DT. However, by using the appropriate threshold, either inside or outside, we were able to mitigate these effects.

We envisage that IVEN can be used for a variety of systems, including early postimplantation mammalian embryos, solid organoids such as gastruloids [44–46], lumenised organoids such as lactating mammary acini [47], as well as small tumours and fragments of tissues. In systems with highly elongated cells, such as neurons, IVEN would require further development to account for the elongated morphology of cells, but through anisotropic thresholding or input of cell boundaries, this could be achieved. However, for the preimplantation embryo and other systems with semiregular cell morphologies and sizes, IVEN provides a robust and objective mathematical approach to analysing cell microenvironments.

Using early mouse embryos, we demonstrated that IVEN can be used as a tool to study the cross talk between reorganisation of the local cell neighbourhood and overall changes in embryo architecture. By studying numbers of neighbours during the period of major architectural changes in preimplantation development, we calculated that during early stages of blastocyst formation, both the median number of neighbours and the mode number of neighbours in the population of mural TE is 5. This increased to 6 at the mid stages of the blastocyst expansion in agreement with previously reported 2D data [22]. This suggests that at this stage, average mural TE cells achieve hexagonal packing that is characteristic of epithelial tissues and long associated with the most energy efficient patterning within biology [48–50].

While TE cells (both polar and mural) had clearly defined modal numbers of neighbours, the most frequently occurring number of inside cell neighbours typically ranged between 9 and 12. The appearance of the neighbourhood distribution histograms during blastocyst development seems to suggest that in a 3D environment, the most frequent number of neighbours is not associated with a single value but can range between several possible values (unlike the honeycomb structure in 2D epithelium). A potential explanation may be that the TE, a group of epithelial cells, represents a stable structure with very little cell movement, thus resulting in neighbourhood distributions with more peaked numbers of neighbours, whereas ICM cells, which are able to change their positions relative to each other and thus exchange neighbours frequently [19,51], may exhibit several equally favourable neighbourhood configurations resulting in several numbers of neighbours with equal frequencies. This observation may therefore suggest that not only the number of neighbours of a cell or groups of cells is important, but also the overall frequency distribution of the neighbourhood. Interestingly, it has been recently reported that high NANOG levels are usually associated with cells with 9 neighbours [21], potentially suggesting that different numbers of neighbours may be in some cases associated with particular transcriptional profiles.

IVEN-driven analyses of ICM cell neighbourhoods suggest that the median number of neighbours for inside cells increases during development, approaching 10 neighbours at the 32-cell and 64-cell stage, subsequently increasing to around 11 cells at the time of implantation. In concordance with the total cell number, the average number of neighbours of inside cells appeared to increase slightly over the blastocyst maturation as suggested in [21], thus potentially facilitating larger neighbourhoods within the ICM to allow for cell competition and ICM organisation occurring during cell specification.

IVEN allowed for detection of much more subtle cell topographies than expected; for instance, at the early 16-cell stage, IVEN identified the existence of outside cells that have only outside neighbours. This finding was consistent with previously published work [7,30,38] and supported the notion that very few inside cells are generated immediately after the 8- to 16-cell division.

Although our data suggest that the overall neighbourhood within the ICM changes little over the blastocyst stages, we only consider "snapshots" of development, the 32-, 64-, and 128-cell stages. It may be that between these key stages, cells are actually more mobile and have more variable environments when considered individually. Further studies including

intermediate stages and specific lineage markers would be necessary to better understand the connections between changes in cell size, cell fate decisions, and development of local neighbourhoods and we believe that IVEN is an ideal tool to facilitate such research.

Changes in cell size could potentially influence cell packing, especially in populations of cells where cell size changes during development. During most of preimplantation development, it has long been known that cells decrease in size with each round of division due to the absence of interphase growth [7,39,52]. The exact developmental stage when mammalian embryos initiate normal somatic cell cycles that include growth phase has not been established until now, although it has been postulated that cleavages continue at least until the 64-cell stage in mouse [7,39,52]. During our neighbourhood analyses using the IVEN pipeline, we observed the decrease in calculated thresholds and median distances between neighbours for each cell, which supported the notion of cell diameter reduction throughout the preimplantation period. At the blastocyst stages, outside cells typically had larger distances between single cells in comparison to inside cells. During this same period, however, it was evident that the distances between neighbours were decreasing with developmental stage. As the exact timing when cell growth is initiated in mammalian development was until now ill-defined, we decided to study the changes in cell diameter over the preimplantation period. Our data suggest that the minimum cell diameter is reached at the time of implantation, E4.5 (128-cell stage), and, after this, cells appear to increase in size. Inner cells appear to display an environment with minimal changes to the overall neighbourhood throughout the whole blastocyst period despite the evident reductions in cell size. This supports the notion that cell size does not directly govern the packing of the ICM and that instead cell properties may play a bigger role in the packing and organisation of the ICM. For example, packing of ICM cells could be influenced by the acquisition of Epi versus PrE identity, as previously suggested in [21], where different cell populations developed differently structured local neighbourhoods.

As IVEN depends only on the coordinates of nuclei/cell centres, it could also be used to compare differences between embryos from different culture conditions. To test how well our pipeline could adapt to a dataset of cultured embryos, we tested and compared KSOM-cultured embryos and FGF4-treated embryos. It was evident that the FGF4-treated embryos, whose ICMs consisted primarily of PrE cells were differently organised when compared to KSOM-cultured embryos. Inside cells in FGF4-treated embryos typically had larger distances between neighbours, alluding to a more dispersed morphology. Interestingly, more dispersed and flattened ICM structures were previously reported after FGF4 treatment in mouse and rabbit embryos [33,34]. To support this further, we were able to perform neighbourhood composition analyses where we calculated the percentage of each inside cell's neighbourhood that was composed of other inside cells. This revealed that there were far fewer inside cells that were able to be entirely surrounded by other inside cells in FGF4-treated embryos, further supporting the notion that after FGF4 treatment, ICMs appear flattened against the TE. Additionally, the neighbourhood distribution of the FGF4-treated embryo displayed a more peaked neighbourhood distribution around 9 and 10 neighbours when compared to KSOM-cultured embryos. We hypothesise that the more peaked distribution of the number of neighbours could be indicative of a more epithelial-like organisation of cells as mentioned previously, and this observation for FGF4-treated embryos could further support this.

## Supporting information

**S1 Fig. Schematic outlining the IVEN pipeline (MATLAB).** Data from IMARIS (or equivalent segmentation program) are output as an Excel file and compiled as outlined in tutorials in order to ensure that all data are imported correctly into IVEN (yellow box). Files are then

selected using the standard file browser window (grey box). The embryo schematic and original cell classifications are generated and displayed to allow for user correction of the automatic cell classification (blue boxes). Channel intensities can be used to assist in correction of the cell classification if analysing confocal images. Method of thresholding of the DT is then chosen by the user, with a variety of options and tunable parameters (green box). After analysis, an output of thresholds used and number of cells analysed is output to the command window (orange box). Finally, an Excel file with all original input data as well as the numbers of neighbours and neighbourhood compositions is output, along with the embryo schematic with final cell classifications (purple boxes). DT, Delaunay triangulation; IVEN, Internal Versus External Neighbourhood.
(TIF)

**S2 Fig. Development of the neighbour distance thresholding approach.** (A) Model embryo showing the effects of the cavity on mural TE cells and their neighbours. Untrue neighbours shown by red arrow, true neighbours shown by green line. (B) Two-dimensional in silico blastocyst with overlaid DT between nuclear centres (blue filled circles). Triangulation boundaries shown between opposing cavity cells (red lines) as well as true neighbours (green lines). Imposed threshold (pink circle) ensures that untrue matches are removed from further analysis. (C) Measurement of distances to all neighbouring cells in an in silico example. Orange lines show cell boundaries, blue circles represent cell/nuclei centres, and purple lines show Euclidean distances between neighbours. (D) Example distributions of distances between neighbours of cells within 32-cell, 64-cell, and 128-cell embryos. Dotted vertical lines show potential threshold boundaries as tuned through the value of $k$. (E) Manual testing of IVEN neighbourhood calculation using different values of $k$. Comparison of manual counts of neighbours with the neighbour counts as calculated using IVEN. (F) Approximate speed tests for the MATLAB and Python versions of IVEN. Approximate running times obtained by including generation of user interfaces with immediate closing of windows. Data underlying this figure can be found on the public GitHub repository https://github.com/jessforsyth/forsyth-et-al-2021. DT, Delaunay triangulation; IVEN, Internal Versus External Neighbourhood; TE, trophectoderm.
(TIF)

**S3 Fig. Examples of low and high background intensity CDX2 data.** (A) Low background intensity staining at the 64-cell stage. (B) High background intensity staining at the 64-cell stage.
(TIF)

**S4 Fig. Comparison of compacted and noncompacted 8-cell stage morulae.** (A) Comparison of the number of neighbours of cells from 8-cell stage compacted and noncompacted morulae show no significant difference. (B) No evident difference between compacted morulae neighbour thresholds and noncompacted thresholds. Data underlying this figure can be found on the public GitHub repository https://github.com/jessforsyth/forsyth-et-al-2021.
(TIF)

**S5 Fig. Outside cell composition analysis used to identify mural TE cells.** (A) Example 16-cell stage embryo with a cell (outlined in red) with only outside cells as neighbours. (B) Corresponding embryo schematic with highlighted cell. Slight elongation of morula captured in embryo schematic. (C) Increase in the number of identified mural TE cells against the size of the embryo (total number of cells). (D) Number of identified mural TE using IVEN against the total cell number at the 32-cell stage. No correlation evident between the number of cells around the 32-cell stage and the number of mural TE cells. (E) Example of a 32-cell

stage embryo with a small cavity with no clear mural TE cells. (F) Corresponding embryo schematic showing no obvious cavity within the sample. (G) Example of a 32-cell stage with a more developed cavity and more obvious mural TE cells. Arrows indicate mural TE cells. (H) Corresponding embryo schematic with clearly visible cavity and mural cells indicated. Data underlying this figure can be found on the public GitHub repository https://github.com/jessforsyth/forsyth-et-al-2021. IVEN, Internal Versus External Neighbourhood; TE, trophectoderm.
(TIF)

**S6 Fig. Cell rounding post-disaggregation and embryo-specific cell diameter comparisons.** (A) Disaggregated cells from developmental stages prior to and around implantation. FM464-X dye used to mark cell membrane, H2B-GFP reporter line used to visualise nuclei. Scale bar, 40 μm. (B) Approach to measure the major and minor axes of disaggregated cells and their nuclei. Scale bar, 40 μm. (C) Cell and embryo numbers analysed. (D) Cell circularity close to one for all stages tested, supporting the assumption of the rounding of cells post-disaggregation. (E–K) Measurements of cell diameter for each embryo within each stage. Data underlying this figure can be found on the public GitHub repository https://github.com/jessforsyth/forsyth-et-al-2021.
(TIF)

**S1 Video. Embryo schematic generated by IVEN for an 8-cell stage embryo.** IVEN, Internal Versus External Neighbourhood.
(MP4)

**S2 Video. Embryo schematic generated by IVEN for a 16-cell stage embryo.** Blue and white points indicative of inside and outside cells, respectively. IVEN, Internal Versus External Neighbourhood.
(MP4)

**S3 Video. Embryo schematic generated by IVEN for a 32-cell stage embryo.** Blue, cyan, and magenta points representative of the inside, polar TE, and mural TE cell groups, respectively as classified through the neighbourhood composition analyses. IVEN, Internal Versus External Neighbourhood; TE, trophectoderm.
(MP4)

**S4 Video. Embryo schematic generated by IVEN for a 64-cell stage embryo.** Blue, cyan, and magenta points representative of the inside, polar TE, and mural TE cell groups, respectively as classified through the neighbourhood composition analyses. IVEN, Internal Versus External Neighbourhood; TE, trophectoderm.
(MP4)

**S5 Video. Embryo schematic generated by IVEN for a 128-cell stage embryo.** Blue, cyan, and magenta points representative of the inside, polar TE, and mural TE cell groups, respectively as classified through the neighbourhood composition analyses. IVEN, Internal Versus External Neighbourhood; TE, trophectoderm.
(MP4)

## Acknowledgments

We would like to thank Nitin Sabherwal, Papalopulu Lab, for sharing materials to assist with embryo disaggregation experiments. We would also like to thank other members (current and past) of the Plusa lab for their discussion about the paper. Finally, we would like to thank Anna

Piliszek, Katarzyna Filimonow, Valerie Kouskoff, Stephen Frankenberg, and Silvia Muñoz-Descalzo for their valuable comments on the manuscript.

## Author Contributions

**Conceptualization:** Jessica E. Forsyth, Berenika Plusa.

**Data curation:** Jessica E. Forsyth, Ali H. Al-Anbaki, Nikkinder Modare, Diego Perez-Cortes, Rowena Seaton Kelly.

**Formal analysis:** Jessica E. Forsyth.

**Funding acquisition:** Jessica E. Forsyth, Ali H. Al-Anbaki, Berenika Plusa.

**Investigation:** Jessica E. Forsyth, Ali H. Al-Anbaki, Roberto de la Fuente, Nikkinder Modare, Diego Perez-Cortes, Isabel Rivera, Rowena Seaton Kelly, Berenika Plusa.

**Methodology:** Jessica E. Forsyth, Simon Cotter, Berenika Plusa.

**Project administration:** Berenika Plusa.

**Resources:** Jessica E. Forsyth, Berenika Plusa.

**Software:** Jessica E. Forsyth, Simon Cotter.

**Supervision:** Simon Cotter, Berenika Plusa.

**Validation:** Jessica E. Forsyth.

**Visualization:** Jessica E. Forsyth.

**Writing – original draft:** Jessica E. Forsyth, Berenika Plusa.

**Writing – review & editing:** Jessica E. Forsyth, Simon Cotter, Berenika Plusa.

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
