## [Editor Report · Decision Letter 0]

1 Apr 2021

Dear Dr Plusa, 

Thank you for submitting your manuscript entitled "IVEN: A quantitative tool to describe 3D cell position and neighbourhood reveals architectural changes in FGF4 treated preimplantation embryos" for consideration as a Methods and Resources by PLOS Biology.

Your manuscript has now been evaluated by the PLOS Biology editorial staff, as well as by an academic editor with relevant expertise, and I am writing to let you know that we would like to send your submission out for external peer review.

Please re-submit your manuscript within two working days, i.e. by Apr 05 2021 11:59PM.

Given the disruptions resulting from the ongoing COVID-19 pandemic, please expect delays in the editorial process. We apologize in advance for any inconvenience caused and will do our best to minimize impact as far as possible.

Kind regards,

Lucas Smith, Ph.D.,

Associate Editor

PLOS Biology

---

## [Decision Letter · Decision Letter 1]

11 May 2021

Dear Dr Plusa,

Thank you very much for submitting your manuscript "IVEN: A quantitative tool to describe 3D cell position and neighbourhood reveals architectural changes in FGF4 treated preimplantation embryos" for consideration as a Methods and Resources at PLOS Biology. Your manuscript has been evaluated by the PLOS Biology editors, an Academic Editor with relevant expertise, and by several independent reviewers.

The reviews are appended below. As you will see, the reviewers think the tool provided here is well constructed, useful, and likely to be widely applicable. However the reviewers have made several suggestions to strengthen the study, which will need to be addressed before we can consider your manuscript for publication at PLOS Biology. Additionally the reviewers have commented that the manuscript is currently a bit too long and could be streamlined. To that end, Reviewer 1 has suggested that figure 5 could be removed from the manuscript. While we agree that editing the manuscript to be more concise is important and will help engage a broader readership, we will leave it to you to decide whether to remove this figure from the manuscript, or not.

In light of the reviews, we are pleased to offer you the opportunity to address the comments from the reviewers in a revised version that we anticipate should not take you very long. We will then assess your revised manuscript and your response to the reviewers' comments and we may consult the reviewers again.

Along with addressing the comments from the reviewers, we also ask that you address the following editorial requests:

1) Financial Disclosure request: Please provide the name of the specific funding agency, and if possible the corresponding grant number and URL, for the scholarship for A.H.A. from the Iraq government. 

2) Ethics request: Please indicate the identification number of the protocol approved by the University of Manchester Animal Welfare and Ethical Review Body.

3) Please also provide a blurb which (if accepted) will be included in our weekly and monthly Electronic Table of Contents, sent out to readers of PLOS Biology, and may be used to promote your article in social media. The blurb should be about 30-40 words long and is subject to editorial changes. It should, without exaggeration, entice people to read your manuscript. It should not be redundant with the title and should not contain acronyms or abbreviations. For examples, view our author guidelines: https://journals.plos.org/plosbiology/s/revising-your-manuscript#loc-blurb

4) Data request: I was unable to find the data used to generate the figures on your GitHUb repository (apologies if I missed it). Please find information on our data sharing policy at PLOS Biology below my signature, and please ensure that your manuscript meets these requirements. As specified in more detail below, we will need you to provide the data underlying each figure in your study, as a supplementary excel file or deposited in a publicly available repository. IMPORTANT: we also need you to reference this data in the figure legends. For example, to each figure legend you might add the following statement: “data underlying this figure can be found in supplementary file S1_data.” You will also need to ensure that this data file contains a legend, and is referenced in your data availability statement. 

We expect to receive your revised manuscript within 1 month. 

Please email us (plosbiology@plos.org) if you have any questions, or would like to request an extension. At this stage, your manuscript remains formally under active consideration at our journal; please notify us by email if you do not intend to submit a revision so that we may end consideration of the manuscript at PLOS Biology.

**IMPORTANT - SUBMITTING YOUR REVISION**

*Resubmission Checklist*

*Published Peer Review*

*PLOS Data Policy*

*Blot and Gel Data Policy*

Sincerely,

Lucas Smith

Associate Editor

PLOS Biology

lsmith@plos.org

DATA POLICY REQUEST:

Figure 1E-F; Figure 2 B-D, F-G; Fig 3A-E; Fig 4 B-E; Figure 5 D-F; Figure S2E-F; Figure S3 C-D; Figure S4A-H

**Please also ensure that figure legends in your manuscript include information on WHERE THE UNDERLYING DATA CAN BE FOUND, and ensure your supplemental data file/s has a legend.

**Please ensure that your Data Statement in the submission system accurately describes where your data can be found.

REVIEWS:

Reviewer #1: Review Forsyth et al. "IVEN: A quantitative tool to describe 3D cell position and neighbourhood reveals architectural changes in FGF4 treated preimplantation embryos"

Paper version April 2nd, 2021 - Review due April 23rd.

The spatial organization of cells in mammalian embryos can be very hard to grasp in 3D. This organization is however crucial, for the position of individual cells in an early embryo govern their fate. Each cell integrates signals from its neighbors, and the presence, absence, numbers and identity of these neighbors is crucial to specify their lineage.

 In this article, Forsyth and colleagues introduce a new bioimage analysis tool called IVEN that helps investigate the identity of cells in a 3D embryonic sample, thanks to determining their neighbors. This software tool is distributed twice, as a MATLAB package and as a Python package. The latter has fewer features than the former, but can harness samples with a larger number of cells better. IVEN requires as input a CSV file that specifies the (X, Y, Z) position of all cells, which can be obtained via a 3rd-party tool. The boundary of the embryo is determined using the convex-hull, and IVEN allows for the manual correction of cells falsely labeled as inner cells by this algorithm. In a second step, IVEN determines cell neighbors by building a neighborhood graph using Delaunay triangulation. To prune aberrant edges that can be created in embryos with cavities (Figure 1C, green inset), IVEN automatically computes a distance threshold for cell-cell distance, above which edges are deleted. IVEN then exports a new CSV file, with the class (inside vs outside) of a cell, the number of its neighbors, and the class of these neighbors. IVEN also uses MATLAB or Python libraries to offer 3D visualization of cell positions. The authors then measure the accuracy of IVEN and illustrate its capabilities for mammalian preimplantation embryos studies. In particular, they study the neighborhood composition in mammalian embryos, and how it is altered in the presence of FGF4. They then measure cell and nuclei diameter along development. 

Although very long, the article is well written even for non embryologists. The software and its code are distributed on a public repository, which includes both the MATLAB and Python version. It also ships example data and excellent tutorials. The code is well written, and well commented. The program is split into several functions, again well documented. IVEN is really meant to be used by other scientists and the authors put a lot of effort into documentation and tutorials. I could run both the MATLAB and Python software successfully, albeit with minor difficulties when installing the required dependencies for the Python implementation (as always with academic Python code…)

 The analysis techniques used in IVEN are similar to e.g. the techniques used in Fischer et al., 2020 [22]. The authors of [22] also used Delaunay triangulation to build a neighborhood graph, with a distance threshold. But as noted in this work, a usable tool widely available was lacking so far. The novelty introduced by IVEN are mainly 1) a classification of inside vs outside cells based on convex-hull followed by manual correction and 2) an automated calculation of a distance threshold to prune aberrant edges in the neighborhood graph. As said above, IVEN keeps its promises and I have only a few major and minor points. 

Major points.

1.

I feel like the last part (Cell diameter decreases with developmental stage until implantation & Nuclei change in size… & Figure 5) does not belong in the paper. It is a beautiful study well done and with meaningful results, but it does not rely on IVEN and does not illustrate its capabilities. Given that the article is already very long and is meant to demonstrate IVEN capabilities, I would suggest moving this part to another article.

2.

IVEN uses the convex-hull to classify cells as inside vs outside. The authors properly measure the performance of this technique (Figure 1E). In the discussion they state that they believe this approach is superior to the ellipsoid fitting approach used in MINS [14]. This point would be much stronger with an actual comparison of the two methods. For instance the study of Figure 1G could be made also with the ellipsoid fitting method.

3.

In IVEN a distance threshold is automatically determined to prune aberrant edges, called in the paper neighbor distance threshold. It is calculated as P75 + 0.5 * IQR. In the third and fourth part (Figure 2G and 4B), the authors use this value as a measure of an embryo spatial organisation. Their conclusions are meaningful but my concern is that this threshold is an intermediate value used in a specific software (IVEN). For instance a researcher that wants to also measure this spatial organisation but without IVEN will have some difficulties relating their metrics to the metrics in Figures 2G and 4D. 

 I would like to suggest instead the use of a distance metric independent from IVEN process. For instance, the authors could define a mean distance to neighbors, defined for one cell as the mean distance from this cell to all its neighbors. Then plot the distribution of this metric for all the cells in one embryo, or take its median to compare from one embryo to another. This metric would carry the same information than the IVEN threshold but would be more readily generalizable.

4.

In the same line, IVEN should offer more spatial features in its output. Right now it outputs for each cell the neighborhood composition, and allows incorporating fluorescence intensity. But it should also incorporate metrics on distance to neighbors, defined for instance as above, to ease compaction studies as is done in Figure 2G and 4D. 

5.

Page 12, paragraph starting from line 458. The authors state that the differences in neighbor distribution in Figure 4E are evident, but I find it is not quite the case. The two distributions comparing control and FGF4-treated embryos have the same shape, the same mode. The differences are very subtle. For them to be relevant we would need to know whether we can trust the number of neighbor metrics with a precision better than 1 cell. Can IVEN reliably assess whether a cell has exactly 12 or 13 neighbors with such a high confidence? 

 This is not discussed in the paper and it is an important point. As stated by the authors, in the assays presented here we determine the number of neighbors via an indirect method (Delaunay triangulation) and not by directly observing contacts with membrane labelling. 

Minor points.

6. 

Page 15 line 607. "Completely open-access". Your code is already open-access. Do you mean "without requiring the purchase of a MATLAB license"? 

Reviewer #2: The manuscript of Forsyth et al., describes the development of a new open source script (compatible for both Matlab and Python) called Internal Versus External Neighbourhood (IVEN) that permits automated classification of the position of individual cells (i.e. on the surface or inside) and their number of neighbouring cells (including distance metrics) within derived 3D microscopic datasets of imaged tissues (predicated on nuclear staining); thus, deriving important cellular architectural/structural information of tissue make up, under native or experimentally perturbed/pathological conditions. Importantly, the script also allows for user interaction to correct any (only limited) erroneous measurements and modulation of defined and tissue relevant parameters. 

The authors, use the preimplantation stages of mouse embryo development (in which they are extremely well versed) as a successful proof of IVEN concept. They show IVEN is able to reliably identify emergent outer and inner cells (based on the number of identified neighbouring cells any given cell, at a given developmental stage, has and thresholding the distance between such neighbours) and sub-categorise trophectoderm (TE) between mural- and polar-TE populations. Moreover, they are able to (for the first time and in exacting detail) conveniently and reliably identify the number of neighbouring cells in emerging ICM, mural- and polar TE cells and gain insight into the tissue architecture of these distinct blastocyst cell lineages. Their IVEN derived analysis also acted as a primer to identifying (again for the first time - in this reviewer's experience) the exact developmental stage reductive (in terms of cell size) cleavage divisions, typical of preimplantation developmental stages, transit to more somatic like divisions preceded by cell growth (i.e. at the E4.5 peri-implantation stage). The authors also revealed the step-wise reduction in nuclear volume throughout preimplantation mouse embryo development. Lastly, the authors demonstrated how IVEN was able to detect and report modulation of blastocyst tissue architecture/structure (within the ICM) in response to exogenous FGF4 addition (known to promote a pan-ICM conversion to differentiating primitive endoderm/PrE cell fate at the expense of the pluripotent epiblast/EPI) that was consistent with transit towards formation of a 'double layer' of TE and PrE epithelia. 

Hence, this reviewer can see how IVEN is especially applicable to studies of preimplantation mouse (and mammalian embryos) and would be applicable to the reanalysis if many existing data sets (including pharmacological inhibitor treatments, clonal RNA-mediated specific gene expression knockdowns or genetic knockout models affecting cell fate or the derivation of the hatching blastocyst). Moreover, as the authors suggest, IVEN is equally applicable to other examples of the study of tissue morphogenesis (e.g. organoids, gastruloids, small tumours etc.) 

In compiling this review, the reviewer acknowledges they were assisted by a senior researcher in their lab (particularly given the manuscript is submitted as a "Methods and Resources Article"). This reviewer found the manuscript highly original and innovative. It is well written, easily followed and will be of interest and utility to both researchers within the mammalian preimplantation developmental field and beyond. As indicated above, the described experiments not only provide a proof of the developed (and widely applicable) IVEN concept, they also provide novel insights into mouse blastocyst tissue morphogenesis. Therefore, this reviewer is quite happy to recommend publication of the manuscript in PLOS Biology but would welcome some redress to few specific points. 

Specific points:

1) This reviewer is curious if the authors, after highlighting the "three fundamental embryo scale architectural changes occurring during the preimplantation period" (first line of the Results section - line 183), conducted IVEN analysis on both non-compacted and compacted 8-cell stage embryos? One asks because it could be expected that the comparison of 'neighbourhood composition' would not significantly change but the 'neighbourhood distance threshold' would reduce.

2) This reviewer is also curious about the observation that 16-cell stage nuclei (in individual cells derived from disaggregated embryos) exhibit a strong degree of sphericalness but the blastocyst stages show much more heterogeneity with substantial populations of ellipsoid nuclei (particularly by E4.5). Do the authors consider the formation of the fluid filled cavity (and potential exertion of mechanical forces, particularly in the epithelial mural-TE and later PrE) a contributing factor? How does the relative distribution of such sphericalness correlate with the IVEN identification of mural-TE (if it is possible to measure nuclear sphericalness in intact blastocysts reliably)? This reviewer also notes from the authors' Discussion section, the observed blastocyst stage increases in the variation of nuclei size and shape is speculated to be related to initiation/progression of cell differentiation/specification. Whilst one agrees this is highly likely, would it not have been relatively easy to stain disaggregated cells for specific blastocyst cell lineage protein markers (e.g. CDX2 for TE or SOX17/GATA4 for PrE etc.) to directly test this? However, this is not necessarily a means to delay publication but would conversely strengthen the manuscript.

3) Similarly, the authors suspect those derived inner cells that only have other neighbouring inner cells may be under "differential signalling regimes" (line 347) that could ultimately effect ICM cell fate. Could they not co-IF stain developing blastocysts for PrE/EPI marker proteins in order to categorise the (emerging) fate of this population? 

4) Relating to Fig.4: Can the authors provide evidence that their +FGF4 treatment regime did not (or indeed did) change overall cell numbers? As cell number changes could contribute to changes in tissue architecture that they observed and may merit further discussion.

5) Regarding the comparison of the authors data with that of Aiken et al., (line 509), this reviewer does not consider the statement fair/valid given one study focussed on cells within in intact embryos and the authors data refer to disaggregated spherical cells. 

6) In relation to Fig. 1F (CDX2 intensity expression analysis in IVEN called inner and outer cell populations): Do the authors agree that inner CDX2 expression is equivalent at the 16-cell and 64-cell stages (as depicted in the figure)? Alternatively, do the reported data indicate IVEN is not so robust in reliably identifying inner and outer cell populations at this stage (and potentially beyond)? 

7) The legend to Fig. 3 lacks reference to panel E) describing neighbourhood frequency distributions at the E4.5/128-cell stage.

Reviewer #3: Morphogenetic changes of embryos are driven by developmental processes at the cellular level including cell differentiation, division and rearrangements, which are tightly coordinated and ultimately determine the success of the embryo development. In the present work, Forsyth et al. introduce IVEN, an interactive pipeline to quantitatively describe changes in the environment of individual cells within growing mammalian embryos. In particular, IVEN discriminates between internal (inner cell mass) and external (trophectoderm) cells using neighbourhood analysis. The analysis is based on Delaunay triangulation, a classical triangulation method used to connect points in space generating a convex hull of the point set, and modified to remove point connections with a length higher than a threshold, either computed in an automated fashion or determined manually. The software is written in MatLab and Python, and clear tutorials are provided for both options, making it a tool with a potential to be widely applicable.

The authors apply the approach to the study of the developing mouse embryo, in which a fluid-filled cavity is formed during trophectoderm maturation. The authors use the IVEN pipeline to classify cells in different clusters based on their relative positions, and further demonstrate that FGF4 treatment causes changes in the organization and distribution of cells within the embryo. Finally, the authors show that the diameter of cells and nuclei is decreasing between the 2-cell stage and E4.5, after which they observe a subtle increase in diameter. This work provides an easy-to-follow pipeline to perform a systematic analysis of cell distribution within embryos. The article is clearly written and the method is well explained. However there are some issues that need to be addressed and we will recommend publication upon addressing these concerns.

Major Concerns 

1. In general the text is very long with unnecessary descriptions and inferences from over analysis of data. 20 pages of text with 5 main figures seem like an overkill. Figure 2D can be seen as an example where the green box shows that inside cells exhibit a sub-population of cells with 100% of their neighbours also inside cells. I am not sure what it means to report values like 0.3%, 1.9% and 2.3% when dealing with stages that consist of such few cells (32, 64 and 128 cells respectively). This is one instance where an "overanalysis" adds to the length of the text without necessarily offering a conclusion that provides any deep biological insights. The work is very interesting and the technical achievement is quite adequate to deserve publication as a crisp and to-the-point text without the need for overselling through long text.

2. Along the lines of comment above, I fail to grasp the point of section starting at line 393. Why using the distance threshold as a proxy of cell packing, when cell positions are available and therefore the density and average distance of cells can be directly computed after IVEN cell classification? A similar question comes for the next section, where cell dispersion in FGF4 treated embryos can be directly computed with cell position and classification.

3. It seems that the two last sections on cell dimensions are disconnected from the rest of the text and from the description of the software. While they provide interesting observations, they fail to integrate within the technical framework of the paper. This connects to point 7: why putting so much effort in justifying a meaning for the threshold parameter?

Minor concerns 

1. While the pipeline is clear and easy to follow and the manual inspection of the data gives high flexibility in choosing the right Delaunay threshold, the time required to process large amounts of data can quickly increase (as shown in Sup Fig 2F). A batch processing module capable of processing several files within the same folder using default parameters would greatly improve the applicability of IVEN to e.g. large time lapse datasets. The final results could still be manually inspected with the standard IVEN pipeline to correct classification errors.

2. Fig.1: how are the cells labeled in panel F: are those cells based on IVEN classification? If so, why are there some inner cells with high cdx2 expression and some outer cells with low cdx2 expression? Are these the cells that are mislabeled as in panel E? What is "ne" in panel E? Most likely it is the number of embryos analyzed and that should be mentioned in the figure legend.

3. Why is convex hull a better way to determine outside cells when compared to ellipsoid fitting? Both methods are not capable of describing complex shapes (as explained in lines 206-210). Is the power of IVEN in the manual curation of the datasets? A discussion on this point would help the reader in understanding the advantages of IVEN over ellipsoid fitting.

4. Missing legend for sup fig 2F

5. Consider moving Fig 3 as Sup Fig, as the main text now jumps from fig2F to fig 3 and back to fig2G.

6. General comment on the distribution plots: it would be better to show the average distribution among all embryos analyzed and the errorbar around the mean, rather than pulling all embryo data together in a single distribution. This would help understand, e.g. in Fig.4E, if the differences in the distribution of control and FGF4-treated embryos are significant.

7. All Supplementary figures are referred as Sx Fig instead of Fig Sx (example lines 195, 224, etc). This should be rectified.

Reviewer #4: This manuscript by Forsyth and colleagues describes the development of an open-source pipeline that can be used to position cells with respect to where they lie in the embryo as well as determine the number of neighbours that each cell has. The authors first validate the use of this pipeline by analysing pre-implantation mouse embryos and find that it accurately can distinguish inside from outside cells. They then use the pipeline to analyse how the neighbourhood of each of these cell populations changes during pre-implantation development as well as in response to FGF4 treatment, that induces cells to form a primitive endoderm fate. Importantly, they find that significant changes in number of neighbours and degree of cell packing correlates with when cell fates are being specified, suggesting that these changes may contribute to lineage decisions. Finally, by analysing the changes in nuclear size that occurs during pre- and early post-implantation development they determine that cell division reduces cell size until the post-implantation period when cell growth is restored. Overall, the manuscript reports a very useful tool to study the relationship between embryo geometry and cell neighbourhood and the authors use it to uncover important new principles governing early mammalian development. 

There are just some minor points that would strengthen the manuscript if they could be addressed.

1. Can the system be used to detect dying cells? If so, what is their neighbourhood?

2. Can the authors comment on if changes in in cell neighbourhood and packing precede when the different cell fate choices occur during pre-implantation development, or if they occur after lineage specification events and are therefore likely a consequence of these events?

---

## [Editor Report · Decision Letter 2]

1 Jul 2021

Dear Dr Plusa,

On behalf of my colleagues and the Academic Editor, Joshua Brickman, I am pleased to say that we can in principle offer to publish your Methods and Resources article, "IVEN: A quantitative tool to describe 3D cell position and neighbourhood reveals architectural changes in FGF4 treated preimplantation embryos" in PLOS Biology, provided you address any remaining formatting and reporting issues. These will be detailed in an email that will follow this letter and that you will usually receive within 2-3 business days, during which time no action is required from you. Please note that we will not be able to formally accept your manuscript and schedule it for publication until you have made the required changes.

Your manuscript was assessed by the Academic Editor and the editorial staff, and we are satisfied by your responses to the reviewer comments and our previous editorial requests. Regarding reviewer 2's point 6, we appreciate the additional analysis you have provided showing that IVEN is effective on images with higher background levels (Response to Reviewer Figure 1). We appreciate that this figure indicates IVEN's utility on non-perfect data, and while we do not think that you should replace Figure 1F with the low background figure, we think it could be added it as a supplemental figure if you think it would be helpful. I wonder if you could add an additional panel showing that the same effects are observed in high background images alone, to help make this point? We will ultimately leave it up to you whether to add this as analysis as a supplemental figure, or not.

PRESS

Sincerely, 

Lucas Smith, Ph.D. 

Senior Editor 

PLOS Biology

lsmith@plos.org